# The Salvador Primary Care Longitudinal Study of Child Development (CohortDICa) Following the Zika Epidemic: Study Protocol

**DOI:** 10.3390/ijerph19052514

**Published:** 2022-02-22

**Authors:** Darci Neves Santos, Tânia Maria de Araújo, Leticia Marques dos Santos, Hannah Kuper, Rosana Aquino, Ismael Henrique Da Silveira, Samilly Silva Miranda, Marcos Pereira, Guilherme Loureiro Werneck

**Affiliations:** 1Instituto de Saúde Coletiva, Universidade Federal da Bahia, Salvador 40110-040, Brazil; aquino@ufba.br (R.A.); ismaelhsilveira@gmail.com (I.H.D.S.); samillymiranda@gmail.com (S.S.M.); 2Núcleo de Epidemiologia, Universidade Estadual de Feira de Santana, Feira de Santana 44036-900, Brazil; araujo.tania@uefs.br; 3Instituto de Humanidades, Artes e Ciências, Universidade Federal da Bahia, Salvador 40170-115, Brazil; marquesleticia@hotmail.com; 4International Centre for Evidence in Disability, London School of Hygiene & Tropical Medicine, London WC1E 7HT, UK; hannah.kuper@lshtm.ac.uk; 5Departamento de Epidemiologia, Instituto de Medicina Social, Universidade do Estado do Rio de Janeiro, Rio de Janeiro 20559-900, Brazil; gwerneck@iesc.ufrj.br; 6Fiocruz Piauí, Teresina 64000-128, Brazil

**Keywords:** cohort studies, Zika virus infection, child, microcephaly, Congenital Zika Syndrome, neurocognitive development, primary health care

## Abstract

This article describes the Salvador Primary Care Longitudinal Study of Child Development (CohortDICa). The exposed group was defined by confirmation of Congenital Zika Syndrome (CZS) diagnosed through computed tomography, magnetic resonance or transfontanellar ultrasound. A random selection of the 169 exposed children led to a subgroup of 120 children who were paired with children from the Live Birth Information System, according to birthdate, residence in the same street or neighborhood, and gestational age, resulting in 115 subjects in the non-exposed group. Following recruitment and before the participants completed 42 months, three measures were applied to assess cognitive, motor, and language performance, corresponding to three home visits. Social characteristics of the families and children, and the neurocognitive development of the children will be compared across the CZS exposed group (*n* = 147), the typical children with no exposure to CZS (*n* = 115) and the STORCH exposed group (Syphilis, Toxoplasma gondii, Rubella, Cytomegalovirus, and Herpes simplex) (*n* = 20). Primary Health Care (PHC) should include long-term care strategies for the care of children and family members, and might benefit from the research, teaching, and extension activities provided in this study. In the face of the consequences of the Zika virus epidemic, an opportunity arose to intervene in the integrated care of child development within PHC, including, on an equal basis, typical children and those with delays or disabilities in the first six years of life.

## 1. Introduction

In April 2015, a recommendation was made for the surveillance of microcephaly and/or central nervous system (CNS) impairments in newborns associated with Zika virus (ZIKV) infection in Brazil, as a result of the ZIKV epidemic in the country [1]. Early detection would support early intervention and support. However, a lack of public health services with qualified teams, capable of monitoring and caring for children with such severe health problems in a country marked by social inequalities, was an important challenge to their developmental follow-up.

Disabled childhood care includes family support strategies and educational devices coordinated with health services within the same territory to allow comprehensive care and early stimulation; yet these were often not adequately established [2,3].

At the time, little was known about the effect of Congenital Zika Syndrome (CZS) on growth, development, and neurological functions in the first years of life [4]. In 2015 and 2016, health professionals in Brazil identified a high frequency of microcephaly and damage to the cerebral cortex—the brain center responsible for functions such as hearing, vision, smell, speech, language, sucking, chewing, swallowing, motricity, and motor coordination [5,6,7]. This finding led the Ministry of Health to declare a National Public Health Emergency in November 2015. Although some findings had indicated the presence of developmental impairments in newborns with these neuroanatomical injuries [6,7,8,9], they were generally based upon one-off measurements of development indicators involving small samples.

Another consideration is that cognitive, motor, and language performance are not only influenced by the level of congenital brain damage, but also depend on contextual factors [7,10,11,12]. The relationship between aspects of the affected child’s neurodevelopment and potential differences in relation to typical development remains an open question. Assessing psychosocial measures, such as the quality of home stimulation and primary caregiver mental health, is therefore relevant to any evaluation of the global impacts of CZS. Primary healthcare (PHC) also plays a potentially important role in the interaction between the individual and their context.

The Salvador Primary Care Longitudinal Study of Child Development (CohortDICa) was undertaken to help fill this knowledge gap. The cohort aimed to identify the spectrum of neurocognitive performance in children affected by CZS, to examine the differences in cognitive, motor, and language performance between exposed and non-exposed groups; and to ascertain the influence of contextual factors on the outcome, in order to plan interventions and train human resources for early childhood care in PHC. The study also provided early stimulation for the caregiver-child dyad and psychosocial support for family members, within PHC.

The study defines Congenital Zika Syndrome (CZS) as the main exposure, and child development as the outcome. To attain these objectives, procedures were designed to select the exposed and non-exposed participants, and to identify instruments to reflect the multidimensional interests of the research. The investigation assessed the children’s cognitive, motor, and language performance; primary caregiver characteristics; the family context and the quality of home stimulation. The administration of longitudinal measures coincided with the window of opportunity for early stimulation [12] and required coordination with primary care health services to provide stimulation for the caregiver-child dyad and psychosocial support for family members.

This article describes these procedures, the steps taken, and the challenges encountered in designing this cohort, in order to support the development of studies aimed at the follow-up of children with congenital impairments.

## 2. Materials and Methods

### 2.1. Study Design

A community prospective cohort study was conducted to assess the effects of the congenital neurological disorders associated with the Zika virus on child development using data from children born between 1 August 2015 and 31 July 2016 in Salvador, Bahia, Brazil, registered on the databases of the Municipal Health Department’s Strategic Information Centre for Health Surveillance (CIEVS/SMS) and the Live Birth Information System (SINASC).

### 2.2. Sample Plan and the Territorial Location of Participants

The study recruited children born between 1 August 2015 and 31 July 2016, when the highest frequency of births with suspected CZS were recorded [13]. In December 2016, the CIEVS/SMS database in Salvador registered 714 notifications of suspected CZS cases—investigations into 433 (60.6%) of these were concluded and 47.1% (204/433) were confirmed as presenting neurological alterations [14]. In November 2017, late results confirmed diagnoses of neurological alterations for a further 22 children born within the study period, providing a total of 226 children eligible for the CohortDICa (184 CZS and 42 STORCH -Syphilis, *Toxoplasma gondii*, Rubella, Cytomegalovirus, and Herpes simplex). For the purposes of analysis, the STORCH group is considered separately, since these infections have deleterious effects on child development, although they appeared to be less severe than CZS. Fifty-seven children were excluded or lost, leading to 147 CZS and 22 STORCH study cases.

In order to enroll participants classified as non-exposed to CZS or STORCH, we obtained 84,776 records from the SINASC database for the years 2015 (46,663) and 2016 (38,113). When we linked these databases, we excluded records of births outside the study period and those of non-residents of Salvador, resulting in 32,360 records as a source for selecting non-exposed participants (Figure 1).

Of the 226 children confirmed with neurological alterations and eligible for the study, 120 were randomly paired for sex, age, area of residence, and gestational age with those registered on the SINASC database, resulting in 117 children considered typical and classified as non-exposed (Figure 2). We could not find matched pairs for three children.

With 343 children identified and mapped for area of residence and proximity to health, education, and social welfare resources in each territory, we contacted the health teams within these sanitary districts, requesting cooperation with the study and assistance in locating subjects. Fifty-seven children were excluded or lost due to: (*n* = 21) refusal to participate, (*n* = 16) address not located, (*n* = 15) moved out of Salvador, (*n* = 3) dangerous access, and (*n* = 2) death, resulting in 286 children for cohort recruitment, of whom 147 had neurological impairments related to CZS, 117 presented typical development, while 22 were STORCH positive children and formed a specific subgroup (Figure 1).There were no age (*p*-value= 0.74) or sex (*p*-value= 0.83) differences between the 57 losses and the 286 baseline participants (Figure 2).

#### 2.2.1. Sample Size Calculation

In order to compare differences in cognitive performance between children exposed to CZS and children who were not, we considered an average cognitive performance score of 94.94 (SD 10.14) for children aged under 42 months [15]. To detect a reduction of at least 5 points in the average cognitive performance score for exposure to CZS, at a 5% significance level and 80% power, we estimated a sample size of 132 subjects—66 in each exposure group. Taking repeated measures into account, a design effect of 1.36 was used to adjust this calculation, resulting in a sample of 180 children, 90 in each exposure group.

#### 2.2.2. Definition of the Groups

The exposed group was defined as confirmation of CZS through diagnosis via computed tomography, magnetic resonance, or transfontanellar ultrasound according to procedures recorded by the CIEVS-SMS. A random selection of the exposed children led to a subgroup of 120 children paired with children from the SINASC database (Figure 2), who were defined as the non-exposed group.

### 2.3. Instruments and Procedures

The set of variables assessed by the instruments/procedures applied at distinct data collection points and the number of participants per exposure category throughout the follow-up are summarized in Table 1.

### 2.4. Definition of Variables

In this cohort, cognitive, motor, and language performance constituted the outcome, with CZS as the main exposure, a permanent attribute measure of these participants over the observed period.

To operationalize the outcome, we adopted two measurement criteria from the Bayley Scale of Infant and Toddler Development (BSID-III)—standard performance score and developmental age (DA). Standard performance was obtained by comparing the child’s score with the instrument’s normative sample [16]. Developmental age (DA) was used as a description of the development level obtained by the child according to their assessment’s positive scores. DA takes account of the gross total of the child’s correct answers for each scale and constitutes an indicator of developmental age in months. Although this is an approximate indicator, it reveals qualitative aspects of the child’s cognitive, motor, and language performance.

#### Covariables

Quality of stimulation in the home: Frequencies and percentages of the score obtained from an application of 45 items of the HOME Inventory were used to calculate environmental risk for the child’s development [17]. The full inventory takes approximately 60 min to complete. It must be carried out at the child’s residence or in another environment in which the child spends most of their time, with the child fully awake, and in the presence of the mother or main caregiver.

Positive psychosocial adaptation: Assessed via the score resulting from 25 affirmative responses to the Resilience Scale, where high values indicate high resilience [18].

Common mental disorders: Measured through the score obtained in dichotomous responses to the 20 items of the Self-Reporting Questionnaire (SRQ-20), with suspected disorder classified through 8 or more positive responses [19].

Depression: measured via the Patient Health Questionnaire (PHQ-9) [20], which refers to depressive symptoms over the previous two weeks, varying from 0 (never) to 3 (almost every day), with a maximum score of 27 points [21].

Other covariables: Parental schooling and occupation, average income, and profile of family group, obtained through a socio-demographic questionnaire drafted by the team.

Data were generated from responses provided by the child’s mother/father or primary caregiver.

### 2.5. Baseline Study Procedures

Between April 2017 and March 2018, researchers made two home visits to establish the study’s baseline. Once located, and having agreed to participate, the families signed an informed consent form approved by the Ethics in Research Committee of the Instituto de Saúde Coletiva/Federal University of Bahia, under number 1,659,107, in accordance with the requirements of Resolution 466/12 of the National Health Council. During the first home visit, we assessed the primary caregiver’s mental health, the quality of psychosocial stimulation in the home environment and the family’s socio-economic profile (Table 1); on average, it took 90 min to perform these procedures. During the second baseline visit, we only applied the BSID-III, in order to obtain the first developmental measure [16,18]; this also lasted an average of 90 min.

### 2.6. Data Collection and Management

The team of home assessment interviewers was made up of health students and psychology professionals, who were responsible for training, ongoing supervision, and the production of reliability measures using the Kappa index and the Intraclass Correlation Coefficient (ICC). The BSID-III and HOME were applied in structured printed formats identified by unique numbering, with data entry in EpiData software (Odense, EpiData association, 2010). For the primary caregiver’s socio-demographic and mental health measures, the instruments were modified to a digital format and inputted into Android mobile devices on the Open Data Kit platform (ODK, 2010). These data were downloaded into masks built in Microsoft Excel 2013 software (Microsoft Corp., Redmond, WA, USA, 2013) and transferred to Stata 14 software (StataCorp LP, College Station, TX, USA, 2015). For every 30 instruments applied, consistency analyses of the applications were performed. The databases corresponding to each instrument were exported to Stata 14 software and saved into a single database using the key variable of record of Live Birth Declaration (in portuguese—Declaração de Nascido Vivo: DNV) for each database.

### 2.7. Data Collection Waves

Eleven months later, in March 2018, we finalized baseline data collection. Between May 2018 and March 2019, we returned to the family homes for the first follow-up (Follow-up 1) to measure cognitive, motor, language, and nutritional outcomes. These data collection procedures ensured that the three assessments were carried out before the children reached 42 months of age. The second follow-up (Follow-up 2) therefore began in February 2019 and was completed in August 2019. When we examined the timings of the baseline and first follow-up procedures, we found that the minimum, median, and maximum time intervals between applications were: 30 days, 1 year and 22 days, 2 years and 6 days, respectively. The intervals between the first and second follow-ups were: 2 months and 5 days (minimum time), 7 months and 14 days (median time), 1 year and 6 months (maximum time). The coefficients of variation for these intervals were 26% (baseline and first follow-up) and 33% (first and second follow-up), indicating homogeneity in the intervals between cohort participant applications. Contact with families was maintained throughout the entire follow-up period via digital communication, principally WhatsApp. One-off hearing and oral health assessments took place between July 2018 and December 2019, by appointment at health centers, and lasted approximately 60 min. In addition to the collection procedures, between July 2018 and December 2019, stimulation activities for the caregiver-child dyad and psychological support for families were provided, in line with the intervention to provide care within PHC.

### 2.8. Early Intervention

Dialogue between the research team and public health professionals supported the proposal for caregiver-child dyad stimulation and psychosocial support for family members in close proximity to their homes. Investments were made in selected PHC units to ensure study participants could access this provision. Quality listening may alleviate the impact on caregiver well-being, leading to improved interactions with the child. Our early stimulation and family listening approach included all research participants, independent of exposure, based on an understanding that the care provided in PHC must include all the dimensions of early childhood development, as well as additional efforts to coordinate care and education. This study is part of the 15 cohorts of children, members of the Zika Brazilian Cohorts Consortium (ZBC Consortium) aimed at a more robust analysis, whose results will inform public health policies related to the ZIKV [29]

## 3. Discussion

### 3.1. Health Services and Recruiting the Cohort

This protocol describes a population-based cohort that was established with participants recruited from Salvador’s Municipal Health CIEVS/SMS and SINASC databases at the peak of the CZS epidemic in Salvador, Bahia, Brazil; three child development home assessments were then carried out.

Within the study period, 226 cases of neurological alteration (184 CZS + 42 STORCH) were confirmed by the CIEVS/SMS, while 117 typical children registered on the SINASC were selected. Of the 343 total subjects eligible for cohort recruitment, 57 cases of neurological alteration were excluded or lost, leaving 169 (147 CZS + 22 STORCH) children eligible for the cohortDICa exposure group, and 117 unexposed typical children, or 286 families in total. Once contact had been made, we were left with 274 subjects with complete baseline assessments (147 CZS; 107 typical; 20 STORCH). In the first follow-up, the response rate was 86.8% (*n* = 238), while in the second it was 81% (*n* = 222) (Figure 1).

These response rates are considered adequate, given the access problems the team of interviewers experienced as a result of social mobility in urban peripheries, community violence, and the amount of time families had available for visits. The exposed participants’ addresses were identified through epidemiological surveillance records and confirmed through the Municipality’s PHC services, enabling recruitment to the CohortDICa.

### 3.2. Multidimensional Assessment of Development

In order to minimize measurement error in the developmental outcomes assessed by the Bayley Scale at each of the three cohort measurements, we measured the intraclass correlation coefficient (ICC) for the functions examined at each point, obtaining a coefficient above 0.86. Substantial and excellent agreement was obtained in at least 95% of the questions assessed at each point [30]. Assessing the contextual exposures also required a technical approach with an estimation of the Kappa reliability statistic for the application of the HOME inventory; this reached an average value of 0.73 (SD 0.23) at baseline.

Collective health work requires evidence about the performance of measurement instruments in order to validate their use in population studies. Assessing the child in their own home, or in exceptional cases in their reference primary healthcare unit, resulted in an experience unlike that of the clinical context of a specialized institution. The home assessment carried out in this study expanded the possible expressions of the development of a child with CZS. This was principally because of the research team’s efforts to properly apply child development theories for children with multiple disabilities [31], and the application of the Bayley Scale guidelines to construct an instrument for sensory and motor facilitation [16,31]. This instrument allowed us to assess the performance of 147 children exposed to multiple disabilities in a population context and provided satisfactory levels of inter-rater reliability.

### 3.3. Study Limitations and Notification Criteria

It was initially understood that the occurrence of microcephaly was the systematic consequence of supposed intrauterine infection. However, given the occurrence of congenital anomalies independent of head circumference, the parameters adopted for diagnosis were considered incorrect. Diagnosis confirmation by the Epidemiological Surveillance System responded to a review of the criteria for the notification of events and the consequent redefinition of the baby characteristics that required notification. The children in this study reflect the changes to the notification criteria adopted during the epidemic, which presented a challenge to the health services’ capacity to detect events. Choosing the epidemiological surveillance database as the original source of sample participants restricted the inclusion of subjects with probable congenital neurological alterations, who were not included in the notification criteria at a given time.

Thus, only newborns whose head circumference was measured in the maternity ward, and who fulfilled the criteria of the Ministry of Health protocol at the time, were able to participate in the study, which assumed the underreporting of children with CZS impairments, as distinct from microcephaly. In-depth assessment by clinical examination, neuroimaging procedures, and laboratory tests were limited to certain cases, once the initial diagnostic criteria were proven to be incorrect [13]. We also note the exclusion of non-residents of the city of Salvador, in addition to the loss of subjects following baseline assessment, which may have influenced our results.

However, the study ensured reliable longitudinal measures for the analysis of development outcomes in children with or without exposure to congenital neurological alterations during the epidemic, sufficient evidence to stimulate measures for the promotion and integrated development of Early Childhood services within PHC.

## 4. Conclusions

With the Zika virus epidemic, an opportunity arose to intervene in the integrated care of child development in PHC, including, on an equal basis, typical children and those with delays or disabilities in the first six years of life.

This opportunity enabled us to apply multidimensional research instruments, work as an interdisciplinary team, and coordinate with PHC, in order to involve families in early stimulation, reduce developmental disadvantages, and promote social inclusion.

PHC is expected to include long-term care strategies for children and family members, and should benefit from our research, teaching, and extension activities (including the residency programme in child development hosted by the CohortDICa study). These benefits will improve services for Early Childhood Growth and Development in coordination with Early Childhood Education, thus maximizing opportunities and quality of life for young children and their families.

## Figures and Tables

**Figure 1 ijerph-19-02514-f001:**
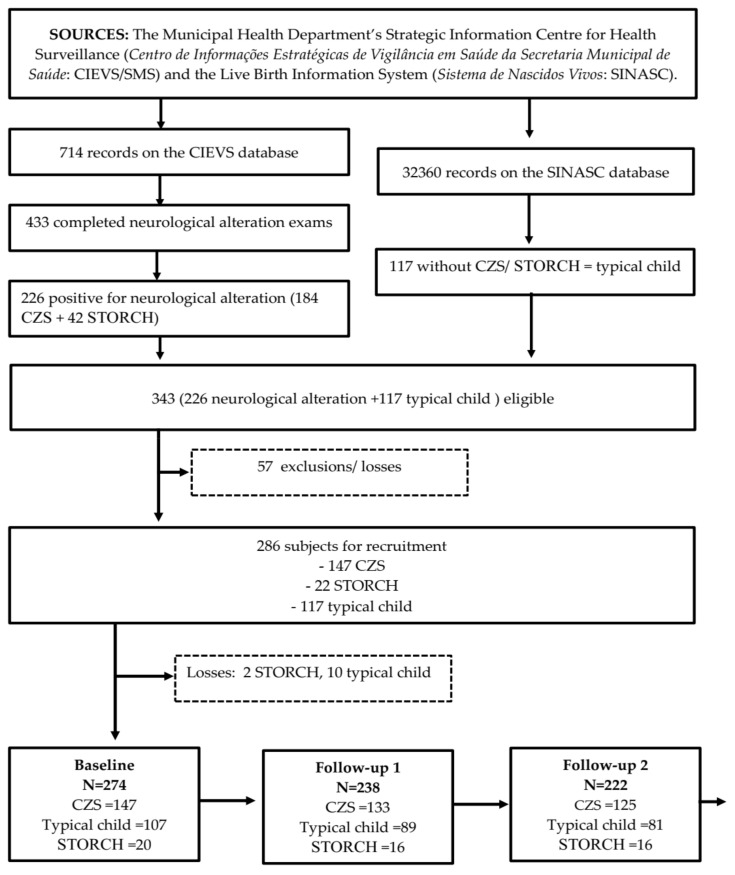
Selection of population study and level of participant involvement. Salvador Primary Care Longitudinal Study of Child Development (CohortDICa).

**Figure 2 ijerph-19-02514-f002:**
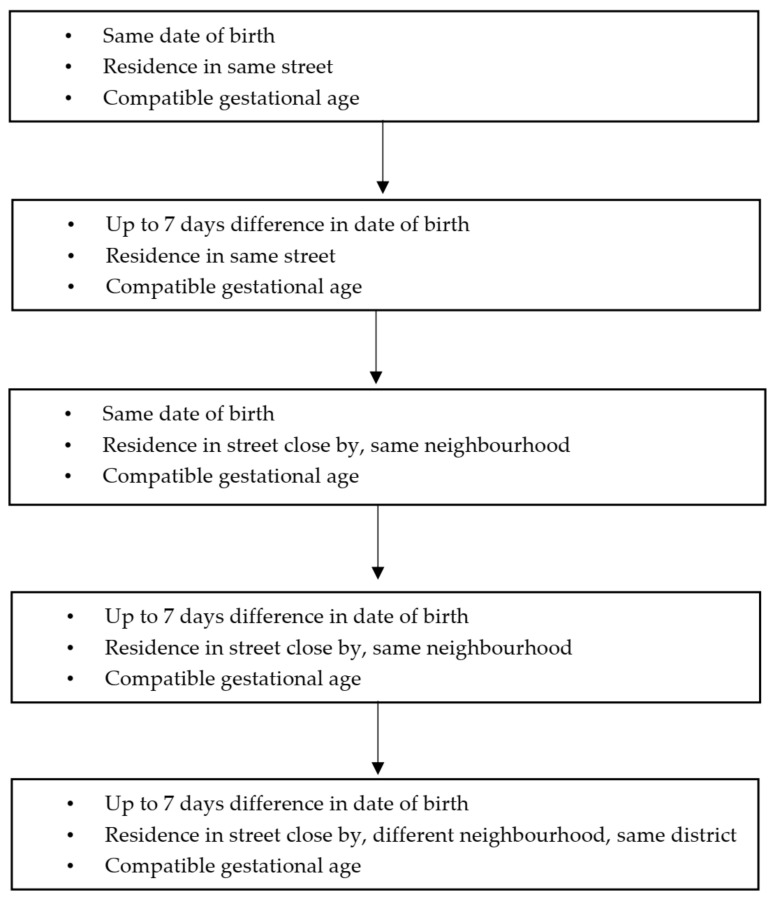
Criteria to select participants without exposure, using records from the SINASC database and pairing with 120 selected children with congenital anomalies 2015–2016, Salvador Bahia.

**Table 1 ijerph-19-02514-t001:** Variables, data collection instruments, duration and assessment points, for the CohortDICa study Salvador, 2017–2019.

Variables	BaselineApril 2017–March 2018(*n* = 274)	Follow-Up 1May 2018–March 2019(*n* = 238)	Follow-Up 2March 2019–August 2019(*n* = 222)	Instruments/Procedures
-Family’s social features, child’s perinatal history and results of tests on the Registry of Public Health Emergencies (42/5.000Resultados de traduçãoRegister of Public Health Emergencies: RESP)	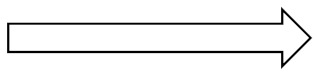			Socio-demographic Questionnaire
-Quality of stimulation in the home and caregiver-child interaction	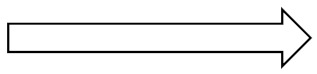			Home Observation for Measurement of the Environment (HOME) Inventory [17].
-Cognitive, motor and language performance	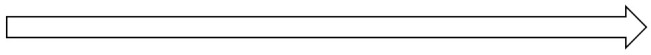	Bayley Scale of Infant and Toddler Development (BSID-III), applied individually [16]
-Maternal Mental Health: Common Mental Disorders and Depressive Symptoms	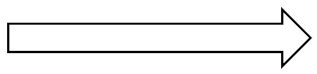		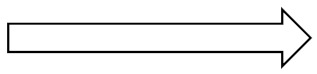	SRQ-20, [19,22]PHQ-9, SANTOS, I. S et al. (2013) [20].
-Positive Personal Psychosocial Adaptation: Personal competenceAcceptance of Self and Life			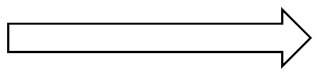	Resilience Scale [18]
-Anthropometric nutritional status, body fat distribution and food consumption		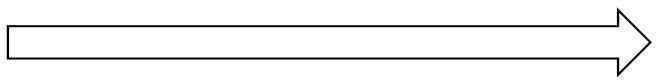	Nutritional assessment indicators, according to the North American Growth in Cerebral Palsy Project [23,24].
-Early signs and symptoms of pediatric dysphagia		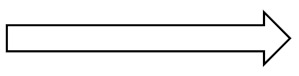		Pediatric Dysphagia Risk Screening Instrument (PDRSI). [25].
-Child functioning:Daily life; Mobility; Cognition/Sociability and Responsibility			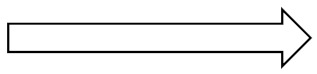	Pediatric Evaluation of Disability Inventory—Computerized Adaptive Testing (PEDI-CAT) [26].
-Health history recorded in previous tests (*n* = 205)		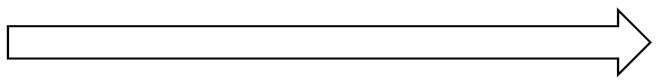	Scanning the child’s booklet and previous tests submitted by the family
-Hearing health (*n* = 28)Peripheral and central hearing, behavioral hearing response to non-calibrated percussion instruments		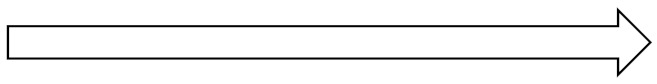	Survey of Evoked Otoacoustic Emissions (AccuScreen OAE device), Immittance testing (otoflex device), Observation of behavioral responses non-calibrated percussion instruments
-Oral health (*n* = 28) Presence of plaque, gingival bleeding and tooth decay		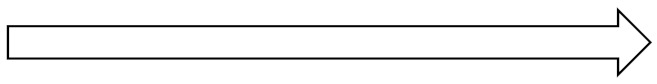	Carter & Barnes flossing index—CBI [27].Assessment Spectrum and Treatment. (CAST) [28]

Baseline (*n* = 274; Exposed: 147; Non-exposed: 107; STORCH: 20), Follow-up 1 (*n* = 238; Exposed: 133; Non-exposed 89; STORCH: 16), Follow-up 2 (*n* = 222; Exposed: 125; Non-exposed: 81; STORCH: 16). Key: period of application for each instrument.

## Data Availability

Not applicable.

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
