# Peer review of "The Salvador Primary Care Longitudinal Study of Child Development (CohortDICa) Following the Zika Epidemic: Study Protocol"

_ijerph, 2022, doi:10.3390/ijerph19052514_

Round 1
Reviewer 1 Report
1. Page 1, line 19: Primary Care Longitudinal Study of Child Development (CoorteDICa): later in the text you use the abbreviation (CohortDICa) for this term. Please correct. The same issue comes up in the legend of Table 1.
2. Page 1, lines 28 and 31: please provide the whole term for the abbreviations STORCH and PHC, before using them for the first time in the abstract.
3. Page 3, paragraph 2.2 and Figure 1: please use the same numbers in your text and in Figure 1, in order to describe the study population. The present format is very confusing. Your audience has to recalculate many times, in order to follow this information between the flow chart (Figure 1) and the text.
4. Page 3, figure 1: what is the meaning of the subgroup STORCH positive children? Were those subjects included or excluded from the analysis?
5. Page 4, line 121: a final sample of 386 children were recruited to the cohort, which is overall 286, according to Figure 1. Please make this point clear (see also comment number 3).
6. Page 6, lines 201-205: please provide the minimum, maximum und median of the follow-up period between the baseline and 1st follow-up, such as between the 1st and 2nd follow-up. Was there a relevant inhomogeneity regarding the follow-up intervals between the study objects?
7. Page 6, table 1: the 2nd follow up period was between 02/2019 and 08/2019, according to the text, which is between 03/2019 and 08/2019, according to table 1. Please correct.
8. Page 6, table 1: which is the meaning of the color bars and the missing gaps in table 1?
9. Please provide the reasons for loss of follow-up. Are they mentioned in the lines 228-229? Please make this point clear.
Author Response
- Page 1, line 19: Primary Care Longitudinal Study of Child Development (CoorteDICa): later in the text you use the abbreviation (CohortDICa) for this term. Please correct. The same issue comes up in the legend of Table 1.
Answer: We appreciate the comments and careful reading of our article. We performed the requested revision, standardizing the term CohortDICa throughout the text.
- Page 1, lines 28 and 31: please provide the whole term for the abbreviations STORCH and PHC, before using them for the first time in the abstract.
Answer: We performed the requested revision in the Abstract ant Text: STORCH (Syphilis, Toxoplasma gondii, Rubella, Cytomegalovirus, and Herpes sim-plex) and, Primary Health Care (PHC).
- Page 3, paragraph 2.2 and Figure 1: please use the same numbers in your text and in Figure 1, in order to describe the study population. The present format is very confusing. Your audience has to recalculate many times, in order to follow this information between the flow chart (Figure 1) and the text.
Answer: We carefully reviewed the information: we standardized the numbers in all text, Figures and Tables, and rewrote the information in the text to ensure clarity.
- Page 3, figure 1: what is the meaning of the subgroup STORCH positive children? Were those subjects included or excluded from the analysis?
Answer: The 20 children whose tested positive for STORCH (Syphilis, Toxoplasma gondii, Rubella, Cytomegalovirus, and Herpes simplex) were identified by laboratory tests registered on the child chart files. The STORCH group was analysed apart because it was composed by cases compatible with CZS, who tested positive for STORCH infections, which are also capable of causing congenital impairments. it has been found that, independent of CZS, presence of STORCH has a deleterious effect on child development, although to a lesser degree than CZS. Even with a sample of only 20 children with some type of STORCH diagnosis, we observed worse living conditions in this group, with mothers with lower education levels, living in more densely populated households.
- Page 4, line 121: a final sample of 386 children were recruited to the cohort, which is overall 286, according to Figure 1. Please make this point clear (see also comment number 3).
Answer: we change the Figure 1.
- Page 6, lines 201-205: please provide the minimum, maximum und median of the follow-up period between the baseline and 1st follow-up, such as between the 1st and 2nd follow-up. Was there a relevant inhomogeneity regarding the follow-up intervals between the study objects?
Answer: The second follow-up (Follow-up 2) started in February 2019 and was completed in August 2019. Taking account of the baseline and first follow-up, the minimum, median and maximum time intervals between applications were: 30 days, 1 year and 22 days, 2 years and 6 days, respectively. For the time interval between the first and second follow-ups, the following results were found; 2 months and 5 days (minimum time), 7 months and 14 days (median time), 1 year and 6 months (maximum time). The intervals had a coefficient of variation of 26% (baseline and first follow-up) and 33% (first and second follow-up), indicating homogeneity in the intervals between cohort participant applications.
- Page 6, table 1: the 2nd follow up period was between 02/2019 and 08/2019, according to the text, which is between 03/2019 and 08/2019, according to table 1. Please correct.
Answer: The selection of population study and level of participant involvement was improved and additional information about the cohort follow-up was insert, in the results.
- Page 6, table 1: which is the meaning of the color bars and the missing gaps in table 1?
Answer: we change the table.
- Please provide the reasons for loss of follow-up. Are they mentioned in the lines 228-229? Please make this point clear.
Answer: In the baseline, of the 343 subjects, 57 were lost due to: refusal to participate (n=21), address not located (n=16), moved out of Salvador (n=15), dangerous access (n=3) and death (n=2), resulting in 286 children for cohort recruitment, of whom 147 had neurological alterations, 117 typical development, and 22 were STORCH positive children. There were no age (p-value= 0.74) or sex (p-value= 0.83) differences between the 57 losses and the 286 baseline participants (Figure 1).

Reviewer 2 Report
The article is very interesting. The proposed objective of describing the procedures and challenges in the design of a cohort, in order to support the development of studies aimed at the follow-up of children with congenital deficiencies. It can be very useful for the design and development of other similar studies.
The introduction frames the study topic very well and the methodology used is impeccable.
Overall, it seems like a good paper, although I suggest that the authors include a final section of conclusions.
Author Response
The article is very interesting. The proposed objective of describing the procedures and challenges in the design of a cohort, in order to support the development of studies aimed at the follow-up of children with congenital deficiencies. It can be very useful for the design and development of other similar studies.
The introduction frames the study topic very well and the methodology used is impeccable.
Overall, it seems like a good paper, although I suggest that the authors include a final section of conclusions.
Answer: we insert the conclusions.

Reviewer 3 Report
The Salvador Primary Care Longitudinal Study of Child Development (CohortDICa) following the Zika epidemic: rationale, objectives, and methods by Darci Neves Santos et al.
This is a longitudinal study (follow up study) of Zika infected newborns versus controls, and additionally an evaluation of the environmental approaches as stimulating ways to improve the outcome of children. The paper requires major revisions.
It is obvious that the study accomplishes with rationale, objectives and methods and the statement in the title is redundant.
Please the Authors to define STORCH and PHC in the abstract (and STORCH in the text and algorithm as well).
Lines 80-87 in the introduction are more compliant with “methods”,
Abbreviations of 2.1 paragraph seems redundant if they are not frequently used in the text, please the Authors to check carefully.
The text is difficult to be read and a deep simplification is needed. Demographics data could be summarized in a table as well as the results, by cutting out the meaningless variables (i.e. street of residence) and by focusing on the main objectives. The colored scheme is not the best way to describe the results of the study. Furthermore the zika virus data in the introduction should be simplified with the main background data on the field, more than with the steps followed in public health (please the Authors to refer to lines 47-55 in introduction).
Author Response
The Salvador Primary Care Longitudinal Study of Child Development (CohortDICa) following the Zika epidemic: rationale, objectives, and methods by Darci Neves Santos et al.
This is a longitudinal study (follow up study) of Zika infected newborns versus controls, and additionally an evaluation of the environmental approaches as stimulating ways to improve the outcome of children. The paper requires major revisions.
It is obvious that the study accomplishes with rationale, objectives and methods and the statement in the title is redundant.
Answer: We change the title
Please the Authors to define STORCH and PHC in the abstract (and STORCH in the text and algorithm as well).
Answer: We insert additional information about STORCH( Syphilis, Toxoplasma gondii, Rubella, Cytomegalovirus, and Herpes simplex)
Lines 80-87 in the introduction are more compliant with “methods”,
Abbreviations of 2.1 paragraph seems redundant if they are not frequently used in the text, please the Authors to check carefully.
Answer – We performed changes in the article.
The text is difficult to be read and a deep simplification is needed. Demographics data could be summarized in a table as well as the results, by cutting out the meaningless variables (i.e. street of residence) and by focusing on the main objectives. The colored scheme is not the best way to describe the results of the study. Furthermore the zika virus data in the introduction should be simplified with the main background data on the field, more than with the steps followed in public health (please the Authors to refer to lines 47-55 in introduction).
Answer – We performed changes to make the text more objective and clear

Round 2
Reviewer 1 Report
Dear Authors,
thank you for providing comprehensive and convincing answers to my questions and queries and made changes, which have improved the quality and increased the publishing potential of your work.
Best Regards
Author Response
English language and style were revised. The article changes are in red
Reviewer 3 Report
The paper is acceptable, as further simplification could be implemented to improve the audience ' comprehension. The Authors should cut out from title the “rationale…. methods”, that are obvious essential parts of a study.
According to the previous revision, there are many variables meaningless across the algorithm.
Please the Authors to carefully check the refs, i.e., n.16
Author Response

(The authors gave the same response as above.)
